# Intranasal Delivery of a Silymarin Loaded Microemulsion for the Effective Treatment of Parkinson’s Disease in Rats: Formulation, Optimization, Characterization, and In Vivo Evaluation

**DOI:** 10.3390/pharmaceutics15020618

**Published:** 2023-02-12

**Authors:** Mohd Imran, Mazen Almehmadi, Ahad Amer Alsaiari, Mehnaz Kamal, Mohammed Kanan Alshammari, Mohammed Omar Alzahrani, Faisal Khaled Almaysari, Abdulrahman Omar Alzahrani, Ahmed Faraj Elkerdasy, Sachin Kumar Singh

**Affiliations:** 1Department of Pharmaceutical Chemistry, Faculty of Pharmacy, Northern Border University, Rafha 91911, Saudi Arabia; 2Department of Clinical Laboratory Sciences Department, College of Applied Medical Sciences, Taif University, Taif 21944, Saudi Arabia; 3Department of Pharmaceutical Chemistry, College of Pharmacy, Prince Sattam Bin Abdulaziz University, Al-Kharj 11942, Saudi Arabia; 4Department of Pharmaceutical Care, Rafha Central Hospital, North Zone, Rafha 76312, Saudi Arabia; 5Faculty of Pharmacy, King Abdulaziz University, Jeddah 42210, Saudi Arabia; 6Faculty of Medicine, King Abdulaziz University, Jeddah 42210, Saudi Arabia; 7Department of Biochemistry and Chemistry of Nutrition, Faculty of Veterinary Medicine, University of Sadat City, Sadat City 32897, Egypt; 8School of Pharmaceutical Sciences, Lovely Professional University, Phagwara 144411, India; 9Faculty of Health, Australian Research Centre in Complementary and Integrative Medicine, University of Technology Sydney, Ultimo, NSW 2007, Australia

**Keywords:** lipophilic drug, microemulsion, mucoadhesion, neurodegenerative disease, central composite design, zeta potential

## Abstract

A mucoadhesive microemulsion of lipophilic silymarin (SLMMME) was developed to treat Parkinson’s disease (PD). Optimization of the SLM microemulsion (ME) was performed using Central Composite Design (CCD). The composition of oil, surfactant, co-surfactant, and water was varied, as per the design, to optimize their ratio and achieve desirable droplet size, zeta potential, and drug loading. The droplet size, zeta potential, and drug loading of optimized SLMME were 61.26 ± 3.65 nm, −24.26 ± 0.2 mV, and 97.28 ± 4.87%, respectively. With the addition of chitosan, the droplet size and zeta potential of the developed ME were both improved considerably. In vitro cell toxicity investigations on a neuroblastoma cell line confirmed that SLMMME was non-toxic and harmless. In comparison to ME and drug solution, mucoadhesive ME had the most flow through sheep nasal mucosa. Further, the in vitro release showed significantly higher drug release, and diffusion of the SLM loaded in MEs than that of the silymarin solution (SLMS). The assessment of behavioral and biochemical parameters, as well as inflammatory markers, showed significant (*p* < 0.05) amelioration in their level, confirming the significant improvement in neuroprotection in rats treated with SLMMME compared to rats treated with naïve SLM.

## 1. Introduction

Parkinson’s disease (PD) is a neurodegenerative disease marked by resting tremors, rigidity, slowness of movement, abnormal gait, and balance problems [1]. PD is the world’s second most prevalent progressive and chronic neurodegenerative condition, marked by the irreversible loss of dopaminergic neurons in the substantia nigra (SN) and their connections to the striatum. As a result, the nigrostriatal pathway’s function deteriorates, leading to the development of movement disorders [2]. The presence of Lewy bodies, which are intraneuronal proteinaceous cytoplasmic aggregates, include α-synuclein, ubiquitin, and neuro-filaments. These are observed in all affected regions of the brain and are well-known as the diagnostic hallmarks of PD [3]. Among every 100,000 people, the incidence of Parkinson’s disease ranges from 8 to 18 cases per year. When over the age of 65, about 1–2% of the population has Parkinson’s disease, and this number rises to 3–5% in those 85 and older [4,5,6].

Glutamate is the primary excitative neurotransmitter in the brain, influencing key physiological processes such as neuronal growth, synaptogenesis, and synaptic plasticity [7,8]. Excessive release of glutamate produces excitotoxicity, which leads to neurodegeneration. Riluzole, fluoxetine, and dexmedetomidine are the most common medicinal neuroprotectants that limit glutamate release, resulting in central nervous system (CNS) depressing effects [9,10]. Whereas, Levodopa, dopamine agonists, monoamine oxidase inhibitors (such as selegiline), amantadine, anticholinergics (such as trihexyphenidyl), carbidopa, and entacapone are some of the medications that are presently used to treat Parkinson’s disease by managing the level of dopamine [11].

However, these treatment modalities cause psychosis, motor difficulties, and obsessive-compulsive disorder [12,13]. As a result, the increased interest in alternative therapeutics for neurodegenerative disorders, such as Parkinson’s disease, has centered on the neuroprotective and antioxidant properties of natural compounds. These may offer options to mainstream therapeutics because of their high efficacy and low side effects. Several bioactive natural compounds, including silymarin (SLM), have taken attention due to their neuroprotective effect and safety [14,15]. SLM is a polyphenolic flavonoid derived from the seeds of milk thistle (Silybum marianum, Family Asteraceae) with silybin as its main component (70–80%). It has been widely used [16,17] in a variety of neurological illnesses, including Alzheimer’s disease, Parkinson’s disease, and cerebral ischemia, among others. Other isomeric flavonolignans include isosilybin, silychristin, and silydianin. SLM provides neuroprotection by reducing oxidative stress, modifying cellular apoptotic machinery, and modulating estrogen receptor machinery [18]. SLM also acts on the CNS by suppressing neuroinflammation, attenuating brain damage, and ameliorating cognitive deficits in various models of neurological disorders [19,20,21]. Despite sharing the immense neurotherapeutic potential of SLM, its clinical applicability has been limited due to poor aqueous solubility (0.04 mg/mL) and oral bioavailability (23–47%) [22]. In addition, SLM suffers from first-pass metabolism. In the past two decades, the nanotechnology approach has been extensively used to increase the drugs’ solubility and permeability. Furthermore, it was reported that nanotechnology has no effect on the inherent properties of medicinal compounds. The current study was designed to develop micro-SLM formulations in order to enhance the therapeutic potential of the drug.

The blood-brain barrier (BBB) continues to be the most difficult barrier to cross for brain medication bioavailability. Efflux transporters, like P-glycoprotein (P-gp), are found in the endothelial cells that make up the BBB. This means that many lipophilic compounds, like potential therapeutic agents, cannot pass through the BBB. Due to the critical importance of effective drug delivery to the brain, a variety of approaches have been evaluated to minimize the effects of the BBB, including the use of prodrugs, inhibiting efflux transporters, disrupting the endothelial tight junctions that, along with the cell membrane, form the physical barrier, and nasal administration.

The high permeability of nasal mucosa and broad surface area allows for a quick onset of the therapeutic effect of the drug. The low metabolic milieu of the snout has the ability to overcome the constraints of the oral route and emulate the advantages of intravenous administration. Furthermore, nasal administration reduces the lag time associated with oral drug delivery and has the fewest negative systemic effects. The ability to target the central nervous system without passing through the BBB is an intriguing advantage of nasal medication administration. Harnessing the edge of intranasal delivery over the conventional route for brain drug delivery, the present study was designed to prepare a mucoadhesive microemulsion (ME) for the nose-to-brain delivery of SLM and seek a better alternative for the treatment of Parkinson’s disease.

Thus, this study is based on a multipronged approach to exploring SLM for its anti-PD potential by converting it into a microemulsion that would reach the brain by penetrating the blood-brain barrier. There are studies wherein microemulsions of SLM have been prepared and reported for different pharmacological actions and through various routes. However, the formulation of a mucoadhesive microemulsion of SLM using a quality-by-design approach that was delivered through the intranasal route has not been reported earlier. This indicates the novelty of the study. Furthermore, the study has proven the development of a product through various in vitro studies and in vivo studies on PD-induced rats.

## 2. Materials and Methods

### 2.1. Materials

For research purposes, Merck Pvt. Ltd. sold SLM to us. Cremophor RH 40 was purchased from BASF Personal Care, while Capryol 90, Labrafac PG, Labrafil M 1944 CS (LMCS), Labrasol, and Transcutol P (TP) were received as gift samples from Gattefosse. Abitec Corporation provided Capmul MCM (Columbus, OH, USA). Tween 80 (T80), Tween 20 (T20), methanol, Span 80, propylene glycol, Span 20, hydrochloric acid, disodium hydrogen phosphate, nitro blue tetrazolium (NBT), orthophosphoric acid, monobasic sodium phosphate, ethanoic acid, dimethyl ketone, L-glutathione reduced, 2-thiobarbituric acid (TBA), sodium chloride, aceto-caustin or trichloroacetic acid (TCA), propylene glycol 400, and N-1-naphthyl ethylenediamine dihydrochloride were acquired from Merck USA. Almond oil, peanut oil, sesame oil, castor oil, and olive oil were purchased from Welch, Holme & Clark Co., Inc. (Newark, NJ, USA). Rotenone was obtained from the Japanese company TCI. Merck (USA) provided L-dopa and lodosyn (carbidopa). Ellman’s Reagent (DTNB), edetic acid (EDTA), and hydrochloride salt of hydroxylamine were procured from Himedia laboratories, Mumbai, India. Edifas B, 1-butyl alcohol, pyridine, disodium carbonate, trisodium citrate, tris-HCI buffer, sodium salt of nitrous acid, and p-aminobenzenesulfonamide were procured from Central Drug House, Mumbai, India. The ELISA kit (Lot no. CB8281) for rat synuclein alpha was bought from Biorbyt (San Francisco, CA, USA). MyBioSource, in the United States, provided an ELISA kit (Lot no. 201908) for rat’s abrineurin or BDNF protein. Raybiotech, Inc. (Peachtree Corners, GA, USA), in the United States, delivered the immunoassay kits for rat TNF-alpha and IL-6. A homogenizer (RQ-127, REMI, Mumbai, India), analytical balance (AX 200; Shimadzu Japan), ultra-fast liquid chromatography (Shimadzu, Kyoto, Japan), UV-spectrophotometer (UV-1800, Shimadzu, Japan), incubator (REMI, India), ELISA plate reader (iMark Microplate Reader, BIORAD, Hercules, CA, USA), actophotometer (INCO Pvt Ltd., Mumbai, India), rotarod apparatus (INCO, Pvt Ltd., India), centrifuge (CM-12 Plus, REMI, India), and pH meter (Phan, Lab India, Mumbai, India) were employed to conduct this research work. SK.N.SH (Human Neuroblastoma Cell Line; ATCC HTB-11) were gifted by the National Centre of Cell Science, Pune, India.

### 2.2. Experimental

#### 2.2.1. RP-HPLC Method Development

The RP-HPLC method reported by Musuluri et al. was used for the estimation of SLM (considering the silybin peak as the major component peak) in the microemulsion using Nucleodur C-18, with a 5 μm column having a 250 × 4.6 mm internal diameter [23]. The mobile phase used was composed of methanol: acetonitrile: tetrahydrofuran in the ratio of 30:65:05 (*v*/*v*/*v*) (pH 5.4), and the flow rate was kept at 0.8 mL/min. The run time was 20 min. The detection wavelength was 285 nm. The retention time of the silybin peak was 7.6 min. The peaks for isosilybin, silychristin, and silydianin were found at 12.7, 13.4, and 14.6 min, respectively, with negligible height for detection. Hence, the silybin peak was taken for quantification and further method validation. In the concentration range of 2–10 g/mL, the technique was determined to be linear with an R^2^ of 0.999. The percentage recovery was found to be between 97 and 99%, and the relative variance among responses was less than 2%.

#### 2.2.2. Selection of Solubilizer

The solubility study was performed for SLM by adding its known excess (50 mg) to various oils viz.: Capryol 90, Labrafac PG, LMCS, Labrasol, almond oil, castor oil, sesame oil, olive oil, peanut oil, Capmul MCM, and surfactants viz. T80, Span 80, T20, Span 20, Cremophor RH 40, and co-surfactants viz. Transcutol P, propylene glycol, and polyethylene glycol 400 (PEG 400) were present in vials made up of glass. The volume of solubilizers was kept at 1 mL. Afterward, the vials were stoppered and subjected to cyclone mixing for 2 min. Then these vials were placed at 37 ± 0.5 °C in a shaking water bath and shaken for a period of 72 h. This was followed by centrifugation of the samples at 10,000× *g* for 15 min and separation of supernatants from the sedimented mass [24,25,26]. The obtained supernatants were diluted using hexane or ethanol and quantified using the developed HPLC method at 220 nm.

#### 2.2.3. Formulation Development

##### Preparation of ME

The ME was developed via a process known as spontaneous nano-emulsification. The oil phase (LMCS), surfactant (T80), and co-surfactant (TP) were blended together to make clear isotropic mixtures and gently titrated with an aqueous phase to get clear and transparent MEs [27].

##### Pseudo-Ternary Phase Diagram (p-TPD) of ME

Based on the results of the solubility studies, the selected oil (LMCS), surfactant (T80), co-surfactant (TP), and water were used to construct the p-TPD. In this case, the Smix was a blend of T80 and TPS. The ratio of the oil phase to the Smix was changed from 1:9 to 1:1, and the Smix was created using water as the aqueous polar/continuous phase. A total of 27 formulation prototypes (F1-27) were prepared. P-TPD was plotted by keeping LMCS, Smix, and water at three different vertices of a triangle, and the observations were in terms of opaqueness and transparency, indicating the formation of the ME or macroemulsion. Todd Thompson Triplot software was used for plotting the p-TPD. Based on the transparent area found in the diagram, the concentration range of solubilizers was further selected to apply DoE [28].

##### Design of Experiment

After the selection of the microemulsion region, the level of oil, Smix, and water was optimized using the design of the experiment. By altering independent factors, such as oil content, surfactant and co-surfactant concentration, and water, a central composite design (CCD) was employed to optimize the formula composition. It assisted in determining the major effects, interactions, and quadratic effects of various formulation constituents on droplet dimension, zeta potential, and drug loading. Design-Expert^®^ software was used to conduct the study [27]. Based on the results of p-TPD, LMCS was varied in the range of 10–30% *v*/*v*, T80 in the range of 15–35% *v*/*v*, TP in the range of 10–20% *v*/*v*, and water in the range of 30–70% *v*/*v*, (Table 1).

##### Preparation of Mucoadhesive ME

By adding a chitosan solution to the optimized batch of SLMME, SLMMME was created (1%, *w*/*v*) [29]. 

#### 2.2.4. Characterization of Optimized Formulations

##### Droplet Size Analysis (DS), PDI, and Zeta Potential (ZP)

The DS, PDI, and ZP of optimized SLMME and SLMMME were measured using a particle size analyzer (Malvern Instruments) by diluting the sample and transferring the solution to separate sample cells that were used for PS and ZP measurement. The sample analysis was carried out as per the procedure mentioned in Beg et al., 2016 and Joshi et al., 2022 [30,31]. 

##### pH, Viscosity, and Refractive Index

The pH of SLMME and SLMMME was checked using a digital pH meter. A Brookfield viscometer (LV model, DVIII ultra programmable rheometer) was used for the determination of the viscosity of SLMME and SLMMME using an S-61 spindle at a temperature of 25 °C. Their refractive index was checked using an Abbe refractometer.

##### Drug Loading

The drug loading (%) was obtained using Equation (1) [24].
(1)$$\mathrm{Drug}\;\mathrm{Loading}\;\mathrm{capacity}=\frac{\mathrm{Weight}\;\mathrm{of}\;\mathrm{the}\;\mathrm{entrapped}\;\mathrm{drug}\;\mathrm{inside}\;\mathrm{the}\;\mathrm{formuation}}{\mathrm{Total}\;\mathrm{weight}\;\mathrm{of}\;\mathrm{formulation}}\times 100$$

##### Thermodynamic Stability Studies (TSS)

TSS were performed for the optimized microemulsion using three different methods. 

Freeze-thaw cycle: After being stored at −20 °C for 24 h, the improved formulation was brought to room temperature. It was visually observed to check for any turbidity or phase separation after returning to its liquid state.Centrifugation test: Following the freeze-thaw cycle, the formulation was centrifuged for 30 min at 3500× *g*.Heating and cooling cycle: The optimized NE was placed at 4 °C in the refrigerator (cooling cycles), whereas, in heating cycles, the formulation was kept at 45 °C. The cycle took 48 h to complete. All samples were evaluated three times [28].

In all cases, phase separation, turbidity, DS, PDI, ZP, and drug loading were examined in the formulation.

##### Cytotoxicity Studies

The blank and SLMMME were tested on a neuroblastoma cell line (SK.N.SH) [32]. In a humid environment containing 5% CO_2_, the cells were placed in a minimal essential medium (MEM), which was enriched with 10% (*v*/*v*) fetal bovine serum, penicillin (100 IU/mL), streptomycin (100 g/mL), and amphotericin B (5 g/mL), at 37 °C until they reached confluency. For the experimental process, the cells were planted into multiwall culture plates. To perform the assay, the suspension was prepared by suspending 5000 cells/well and added to a total of 96 wells. The cytotoxicity was tested at various concentrations of SLMMME and SLMME (10, 20, 40, and 80 µg/mL). The cells suspended in MEM without SLMME were used as the control. In a humid incubator with 5% CO_2_, the microtiter plates loaded with cell suspension were incubated at 37 °C and for 72 h. Doxil (doxorubicin) was used as a positive control compound. Daily morphological changes in the cells were examined for microscopically observable modifications, such as loss of monolayer, granulation, and cytoplasmic vacuolation. It was discovered that there was a cytopathic effect. The SRB (Sulforhodamine B) assay was used to determine the GI50 (drug’s amount causing 50% growth suppression of the cells), TGI (drug’s amount causing total suppression of the cells), and LC50 (drug’s amount causing 50% cell death) [27].

##### In Vitro Drug Release Studies

SLMME, SLM mucoadhesive ME (SLMMME), and SLMS were studied in vitro in modeled nasal fluid containing 1% SLS utilizing the dialysis bag technique [33] for the first 2 h. A dialysis membrane (Dialysis membrane-150, HiMedia, Mumbai, India) with a pore size of 2.4 nm and a molecular weight limit of 12,000–14,000 Da was employed. The dialysis bag contained 1 mL of formulation having an SLM concentration equivalent to 10 mg/mL. Anhydrous sodium phosphate monobasic (7.5 mM), anhydrous sodium phosphate dibasic (3 mM), halite (NaCl) (150 mM), sylvite (KCl) (40 mM), and calcium chloride were all present in the simulated nasal fluid (5 mM). At the end of 2 h, the dialysis bag was placed in brain simulated environment (pH 7.4), and the study was continued for another 10 h. Overall the study was conducted for 12 h. For the 12 h leading up to use, the bags were pre-soaked in distilled water. The dialysis bags were filled with the formulations and drug suspension, which were then sealed on both ends. The bags were submerged in 500 mL of dissolving media at 37 ± 0.5 °C for 30 min. Samples were collected at periodic intervals and replenished with the same volume of fresh dissolving medium in order to preserve a consistent volume. HPLC was used to evaluate the samples. All of the measurements were performed three times. One-way ANOVA was used to examine the release data, with Bonferroni as a post hoc test.

The data obtained from the In vitro release studies of SLMME and SLMMME were fitted to various kinetic models, such as zero order, first order, Higuchi model, Hixon Crowell, Weibull, and Korsmeyer Peppas models. The mechanism and kinetics of drug release were determined by the obtained correlation coefficient (R^2^) [34]. The model showing the highest value of R^2^ was considered.

##### Ex Vivo Diffusion Studies

The ex vivo diffusion experiments of SLMME, SLMMME, and SLMS (SLMS) were carried out using freshly isolated sheep nasal mucosa obtained from a slaughterhouse and placed in a buffer solution (PBS-pH 6.4) [35]. The nasal membrane was gently removed and freed of any attached tissues. The study used tissues with a thickness of 0.2 mm. The nasal membrane that had been cut out was placed on a Franz diffusion cell. These cells were obtained from Trover, Nakodar, India, and the surface area was 1.79 cm^2^ with a volume of 25 mL. For 30 min, the tissue was stabilized in both compartments using mimicked nasal fluid with magnetic stirring. After 30 min, both compartments’ solutions were emptied and replaced with new SNF. About 1 mL of drug solution was injected into the sac made of sheep nasal mucosa (donor compartment), and the study was started. At regular time intervals, the samples were withdrawn from the receptor compartment, filtered (via a 0.45 µm membrane filter), and injected into HPLC to measure the SLM’s concentration. Similarly, diffusion studies were conducted for SLMME and SLMMME. The withdrawn media of the receptor compartment during sample withdrawal were replaced with fresh medium. The study was conducted for 12 h. The data to assess the permeation of SLM from different formulations was plotted between the amount of drug penetrated per unit mucosa surface area (µg/cm^2^) versus time (h). Using linear regression analysis, the steady-state flow (Jss, µg/cm^2^ h) was estimated from the slope of the linear component of the graph. All of the measurements were performed three times. The permeation data was examined by 1-way ANOVA employing Bonferroni as a posttest.

#### 2.2.5. Animal Study

The study was carried out under the approved protocol number VUSC-005-1-22 dated 13 April 2022. All the protocols used were as per the guidelines of the Institutional Animal Care and Use Committee (IACUC), Faculty of Veterinary Medicine, University of Sadat City, Egypt. Albino Wistar male rats aged between 7–8 weeks and weighing between 250–300 g were included in the study. Polypropylene cages layered with husks were used to keep the rats. The beddings were changed at an interval of 1 day. A light and dark cycle of 12 h each, a temperature of 25 ± 2 °C, and a relative humidity of 55 ± 10% were maintained throughout the study. A standard pellet diet was provided to them with free access to water. 

##### Parkinsonism Animal Model

The rats were distributed into eleven batches, with a total of six rats in each batch. Rotenone (2 mg/kg) was suspended in sunflower oil and administered subcutaneously to treatment groups. The control group received a mixture of sunflower oil and normal saline. In 0.5% CMC, L-dopa (100 mg/kg) and lodosyn (carbidopa) (25 mg/kg) were utilized as conventional treatment medicines. SLM (High Dose), SLMME (5 mg/kg and 10 mg/kg), and SLMMME (5 mg/kg and 10 mg/kg) were administered for 35 days to rats through the intranasal route. Assessment of behavioral alterations was performed prior to starting dosing and at intervals of one week up to the 35th day. Animals were sacrificed after treatment and behavioral analysis, and the midbrain was extracted for future study. The detailed protocol is shown in Table 2.

#### 2.2.6. Behavioral Analysis

##### Locomotor

The rats’ locomotor behavior was measured using an actophotometer. The actophotometer was made up of a square metallic container with a lid, similar to an activity cage. The activity cage was constructed using infrared photocell beams spanning the frame’s axis. The instrument recorded the number of beams crossed (successive intermission of one beam followed by disruption of an adjacent beam) for 10 min, and this was utilized as an amount of spontaneous movement [36].

##### Muscle Coordination (MC)

Assessment of MC for all groups was performed using a rotarod, wherein the time acquired for rats staying on the rod was recorded. The rotating rod was kept at a height of 20 cm from its enclosure’s bottom and rotated at a constant speed (25 rpm). The rats were trained to stay on the rotarod prior to the start of this study. The time (in seconds) spent by rats on the rotating rod was recorded as “latency to fall”, which was used as the endpoint for this study. The maximum time limit was set at 120 s [36].

##### Catalepsy

The bar test was utilized to assess catalepsy. A horizontal bar was taken, and rats were placed on the bar in a half-rearing posture. The height of 9 cm from the base of the bar was maintained for this study. The time taken for the removal of a paw from the bar by the rats was noted using a stopwatch with a cut-off time span of 180 s [26].

#### 2.2.7. Biochemical Parameters

At the end of the study, the rats were sacrificed, and their midbrain was isolated. It was suspended in 0.1 M phosphate buffer (pH 7), and the suspension was homogenized at 10,000× *g* for 10 min at a temperature of 4 °C. The supernatant was collected and stored at −20 °C. The homogenized brain was used for the assessment of various oxidative parameters through ELISA kits.

##### Thiobarbituric Acid Reactive Substances (TBARS) Assay

Estimation of TBARS was performed by quantification of malondialdehyde (MDA) level. A 1:1 mixture of tissue supernatant and Tris HCl, pH 7.4 (0.2 mL each), were mixed and incubated at 37 °C for 2 h. This was followed by the addition of TCA (10% ice-cold), centrifugation for 10 min at 1000× *g,* and the addition of TBA (0.67%) in the supernatant. It was heated for 10 min, cooled, and diluted with 1 mL of distilled water. Then, absorbance was recorded at 532 nm. A further computation was performed using MDA’s extinction coefficient, i.e., 0.156 µM^−1^, and the final concentration was reported in nanomoles of MDA per mg of protein [26].

##### Nitrite Assay

For the measurement of nitrite’s production in the homogenate, it was treated with Griess reagent as per the method described by [37]. The composition of Griess reagent included phosphoric acid (5% *w*/*v* in water), sulfanilamide (1 g), and N-1-naphthyl ethylene diamine dihydrochloride (100 mg). The tissue homogenate and Griess reagent were mixed in equal proportion for 5 min and kept in the dark for 10 min for incubation. The absorbance of the suspension was recorded at 540 nm. The concentration of nitrite produced during the study was quantified through a calibration curve method that was prepared between 10 to 100 µM [26].

##### GSH Assay

The technique of Beutler was used to measure the level of GSH [38,39]. The supernatant of tissue homogenate and 10% TCA (1 mL each) were blended in water with 1 mL TCA. The entire solution was centrifuged at 1000× *g* for 10 min. To the supernatant (0.5 mL), 0.3 M disodium hydrogen phosphate (2 mL) and DTNB (0.25 mL, 0.001 M in 1% *w*/*v* sodium citrate) were added, and absorbance was recorded at 412 nm using a UV-spectrophotometer. The concentration of GSH was measured from the standard curve (10–100 µM) of the reduced form of glutathione [26].

##### Catalase (CAT) Assay

CAT assay was performed by the thaddition of tissue supernatant (0.05 mL) and phosphate buffer (1.95 mL, 50 mM, pH 7.0). To the above solution, hydrogen peroxide (30 mM, 1 mL) was added. The optical density (OD) was recorded at intervals of 15- and 30-s using a UV-Visible spectrophotometer at 240 nm. Equation (2) was used to calculate the CAT assay [26].
(2)$$\mathrm{CAT}=\{[(2.3\times \log\frac{\mathrm{OD}\;\mathrm{initial}}{\mathrm{OD}\;\mathrm{final}})÷\Delta t\times 100]÷0.693\}\;/\mathrm{mg}\;\mathrm{of}\;\mathrm{protein}\;$$

Note: “Δt” stands for the time interval at which absorbance was taken (i.e., 15 s).

##### Superoxide Dismutase (SOD) Assay

For measuring SOD, the tissue homogenate’s supernatant (0.1 mL), 2 mL EDTA solution (0.1 mM), NBT 96 mM, sodium carbonate 50 mM (pH 10.8), and hydroxylamine hydrochloride (0.1 mL, 20 mM, pH 6) were added and mixed well, and the optical density was measured at 560 nm for 2 min at an interval of 60 s [26]. The SOD level was calculated using Equation (3).
(3)$$\mathrm{SOD}\;=\left(\frac{\;\Delta \mathrm{OD}\;\mathrm{of}\;\mathrm{control}\;-\Delta \mathrm{OD}\;\mathrm{of}\;\mathrm{sample}\;}{\Delta \mathrm{OD}\;\mathrm{of}\;\mathrm{control}}\;\right)\times \frac{\frac{100}{50}}{\mathrm{volume}}\mathrm{of}\;\mathrm{homogenate}/\mathrm{mg}\;\mathrm{of}\;\mathrm{protein}$$

Here, “ΔOD” stands for the change in absorbance of the control and the sample at 560 nm.

##### Total Protein

The method of Lowry was used to measure total protein, wherein, Folin’s phenol reagent was mixed with tissue homogenate, and changes in color and absorbance were recorded at 750 nm [40]. The calculation of bovine serum albumin was performed using the calibration curve method.

##### Enzyme-Linked Immunosorbent Assay (ELISA)

The levels of alfa synuclein, BDNF, TNF-α, and IL-6 in the rats’ midbrains were estimated using ELISA assay kits. To perform this, midbrain tissues were isolated, homogenated, and processed as per the procedure directed in the kit’s user manual. The homogenates of the test and standard groups were added to the wells precoated with the specific antibody and incubated to activate proteins for binding with the immobilized antibody. After activation, the kits were washed, and the markers (biotinylated anti-Rat Alfa synuclein, BDNF, TNF-α, and IL-6) were added in a sequential manner. The kits were incubated for some time for successful interaction of the tissues with the markers and then washed. After washing, HRP-conjugate and TMB substrate were added. The appearance of a blue color showed the progress of the reaction. This was followed by the appearance of a yellow color upon the addition of the stop solution [26]. The measurement of the yellow color at 450 nm for various groups was performed using an ELISA plate reader (iMark Microplate Reader, Company-BIORAD). The concentration of these biomarkers was dependent on the intensity of color.

##### Dopamine (DA)

The level of DA present in the left hemisphere striatum, prefrontal cortex, and dorsal hippocampus was quantified using a dopamine-based ELISA kit. The procedure was performed as per the user manual provided in the kit. On day 1, the extraction of tissues from test and standard groups was performed, and on the next day, their optical density was measured at 450 nm. The brain tissues were taken in Eppendorf tubes and treated with EDTA-HCl buffer (0.5 mL), sonicated for homogenization, and centrifuged at 1287× *g* for 10 min at 4 °C. The supernatant of all the groups was added to the wells of the extraction plates. The supernatant (20 μL) was transferred into a fresh Eppendorf tube. Additionally, 0.5 mL of each sample was also pipetted into the remaining extraction plate wells and diluted with 0.5 mL of bi-distilled water to correct the volume differences. To all of these, 0.05 mL of release buffer was added to the extracted samples. These samples were treated with 0.075 mL of COMT enzyme solution. A triplicate study was performed, and their optical density was measured using a microtiter plate reader (SPECTROstar Nano, BMG LABTECH GmbH, Ortenberg, Germany) set at 405 nm within 10 min of adding the stop solution [26].

##### Statistical Analysis

The study was performed in replicates, and their mean data with standard deviation were recorded. Further, the results of the behavioral parameters related to all groups were compared using a 2-way analysis of variance (ANOVA). The results of biochemistry and biomarkers of all groups were compared using 1-way ANOVA. All the experimental data were expressed as mean ± S.E.M (standard error mean). The Tukey test was applied to both types of ANOVA. Sigma Stat software version 3.5 was used to develop graphs. The results were considered significant upon achieving a “*p*” value of less than 0.05.

## 3. Results and Discussion

### 3.1. Solubility Studies

The solubility studies were carried out using a series of oils, surfactants, and co-surfactants. It was observed that SLM was maximumly soluble in LMCS, followed by CMCM among oils, T80 among surfactants, and TP among co-surfactants. Hence, LMCS, T80, and TP were used as the oil, surfactant, and co-surfactant, respectively. The solubility of SLM in LMCS, T80, and TP was found to be 15.22 ± 1.56 mg/mL, 22.13 ± 2.14 mg/mL, and 11.13 ± 1.44 mg/mL, respectively. The results are shown in Figure 1.

### 3.2. Pseudo-Ternary Phase Diagram

The pseudo-ternary phase diagram was constructed to find out the zone in which a suitable composition of oil, surfactant, and co-surfactant could result in a clear and transparent emulsion indicative of ME. A total of 27 emulsion prototypes were formulated, but only three formulations, F1, F10, and F19, showed a clear and transparent emulsion (Figure 2). Formulations F2, F3, F4, F9, F11, F12, F18, F20, and F21 showed a translucent emulsion. Based on the obtained results, it was observed that a ratio of oil (LMCS) in the range of 10–30%, surfactant (T80) in the range of 15–35%, and co-surfactant (TP) in the range of 10–20% was found to produce a clear and transparent ME. Hence, CCD was applied in this range to investigate the effect of these solubilizers on droplet size, zeta potential, and drug loading.

### 3.3. CCD

As mentioned above, a randomized central composite design was applied to investigate the effect of varying compositions of solubilizers on various responses, viz., droplet size, zeta potential, and drug loading. A four-factor, three-level design was used with a total of 21 runs (Table 3) with ±α values in each case. The ratio of LMCS varied between 10–30% with a +α value of 36.8179 and a −α value of 3.18. Similarly, the ratio of T80 varied between 15–35% with a +α value of 41.8179 and a −α value of 8.18,207. The ratio of TP varied between 10–20% with a +α value of 23.409 and a −α value of 6.59,104. The ratio of water was varied between 30–70% with a +α value of 83.6359 and a −α value of 16.3641. The lowest value of droplet size was observed in run 15 (46.78 nm), and the largest droplet size was observed in run 10 (152.71 nm). The highest negative value of zeta potential was observed in run 13 (−47.7 mV), and the lowest negative value of zeta potential was in run 4 (−15.91 mV). The highest value of drug loading was observed in run 14 (99.12%), and the lowest value of drug loading was in run 7 (57.35%). Furthermore, ANOVA was applied to check the interaction between responses and variables and found linear, as well as significant (*p* < 0.05) in all the cases, indicating the suitability of the model (Table 4).

The polynomial equations obtained for the responses are depicted in Equations (3)–(5) for (Y1), (Y2), and (Y3), respectively. Equations (3) and (5) represent linear models, and Equation (4) represents a quadratic model.
(4)Y1 [Droplet size]=+93.36−13.61×A−4.60×B+2.32×C−22.36×D 
(5)Y2 [Zeta potential]=−20.76−8.44×A−3.35×B−4.23×C−4.90×D−5.01×AB−6.75×AC−5.37×AD−0.6662×BC
(6)Y3[Drug loading]=+82.91−2.66×A−8.52×B+2.39×C+5.32×D

In the equation, the positive sign before factors indicates the synergistic effect of that factor on the response, whereas the negative sign indicates the antagonistic effect of that factor against that response [41]. In the case of droplet size, it was observed that LMCS, T80, and water have an antagonistic effect on droplet size, whereas TP has a synergistic effect on droplet size. This revealed that LMCS and T80 were important in lowering the emulsion droplet size. This negative effect could be due to the fact that SLM showed good solubility in LMCS and Tween 80. Furthermore, they were found more helpful in breaking the intermolecular forces of SLM, which would have helped in the reduction of their droplet size. In the case of zeta potential, all factors helped in increasing the negative zeta potential of the formulation. In the case of drug loading, TP helped in increasing the drug loading. TP played a multifaceted role in the optimization of the formulation by providing good repulsive forces between droplets by creating a high zeta potential. In addition, it also offered good solubility to SLM. Perturbation plots for responses are represented in Figure 3. The bend curves indicate the intensity of the impact of that variable on the response. In all responses, all factors showed sharp bends indicating their significant effect on the responses. The polynomial equation also helped in the generation of 3D response surface plots that further showed similar observations as indicated by the polynomial equation. The images are shown in Figure 4, Figure 5 and Figure 6a.

The formulation parameters (X_1_, X_2_, X_3_, and X_4_) for SLMME were optimized through the graphical optimization method based on their effect on responses. The constraints for X1 (LMCS) and X2 (Tween 80) were kept at the “minimize” level. For variable X3 (Transcutol P), it was kept at “in range i.e., intermediate level”, and X4 (water) at the “maximize” level. Similarly, the constraints for response Y1 (DS) were kept at the “minimize” level, whereas for response Y2 (ZP) and Y3 (DL), it was at the “maximize” level. The predicted values for response Y1 was in the range of 26.17 to 92.92 nm, Y2 (ZP) was in the range of −12.43 to −34.09 mV, and Y3 was 78.28 to 115.27%. The design predicted the percentage of X_1_, X_2_, X_3_, and X_4_ in the range of 10%, 15%, 10%, and 70% *v*/*v*, respectively. Figure 6b depicts the overlay plot indicating the desirable values of factors and responses.

For validating the predicted value, the experiment was run in triplicate using the suggested values of factors (X) and provided a droplet size (Y1) of 61.26 ± 3.65 nm, zeta potential (Y2) of −33.32 ± 0.4 mV, and drug loading (Y3) of 97.28 ± 4.87%. There was a non-significant difference between the observed values and the predicted value. The optimized formulation passed the tests of freeze-thaw cycle, centrifugation stress, and heating-cooling cycles, as there was an absence of phase separation and drug precipitation. Furthermore, during the freeze-thaw cycle, centrifugation stress, and heating-cooling cycles, the DS was found to be in the range of 63–66 nm, ZP was found in the range of −32 to −35 mV, and drug loading in the range of 96 to 97%. All these values were found to be in close agreement with the optimized results of DS, ZP, and drug loading, as discussed above. After mucoadhesion, the droplet size and zeta potential were increased, whereas drug loading was decreased owing to the mucoadhesive properties of chitosan. The droplet size of SLMMME was 72.34 ± 4.32 nm, the zeta potential was −24.26 ± 0.2 mV, and drug loading was 96.31 ± 5.22%. A significant increase in DS was observed that indicated chitosan coating of the microemulsion. In addition, the ZP was significantly decreased due to the surface modification of the microemulsion by positively charged chitosan.

### 3.4. Cytotoxicity Studies

It is important to note that the lethality of a microparticulate-based formulation, especially for brain distribution, has vital benchmarks to evaluate its safe use. The cell toxicity screenings of optimized SLMMME were performed on SK.N.SH cell lines. For blank ME, SLMME, and SLMMME (at all concentrations), GI50, TGI, and LC50 were found to be greater than 80 µg/mL, as less than a 15% inhibition in cell growth was observed in all cases up to 80 µg/mL. For doxorubicin, the LC50 and TGI values were 48.2 µg/mL and 10.9 µg/mL, respectively. The percent growth curve of SLMMME and SLMME were plotted, and the results showed no inhibition of cell growth at a concentration range of 10–80 µg/mL. The results are shown in Figure 7. The results suggest that the ME did not cause any inhibition of growth and is found to be non-toxic to the cells. Hence, it is considered safe for brain delivery [27].

### 3.5. In Vitro Release Tests

When compared to SLMS, the release of SLM from ME was considerably substantial (*p* < 0.001). Figure 8 depicts the results. SLM is a lipid-soluble drug with a log P value of 1.22 and low aqueous solubility [42]. The results indicated only 2 ± 0.55% of the drug released from SLMS after 2 h, which was considerably poorer (*p* < 0.001) than SLMME and SLMMME. SLMME and SLMMME showed 20.32 ± 1.46% and 6.56 ± 0.52% of release after 2 h in simulated nasal fluid. When the study was continued in brain-simulated fluid (for an additional 10 h), then at the end of 12 h, a significant enhancement in the release of SLM was observed from SLMME (91.67 ± 5.89%) and SLMMME (66.28 ± 4.92%) as compared to SLMS (28.34 ± 4.22%). The sink condition for SLM in SLMS, SLMME, and SLMMME was calculated using the Cs/Cd formula (the ratio of saturation solubility to the maximum concentration of drug in a given volume of medium). The value of Cs/Cd for SLM in the case of SLMS was 0.14, whereas for SLMME, it was 0.92, and for SLMMME, it was 0.63. The higher value of the sink condition indicated enhancement in the sink condition of SLM upon loading it into the ME.

Greater release from ME was attributed to tiny globule size, which increased the contact surface for dissolution [26]. The addition of a mucoadhesive ingredient to ME substantially reduced the release of SLM. This might be attributed to the increased viscosity of mucoadhesive ME, which may act as an obstacle to drug release. Between 4 and 12 h, the release of SLM from SLMME was substantially higher (*p* < 0.001) than that of SLMMME. The release data of ME was fitted into Higuchi’s equations, zero order, and first order. The r^2^ value for Higuchi’s model was found to be higher as compared to any other models (Table 5). If the amount of medication released from a matrix system is directly proportional to the square root of time, it is considered to follow Higuchi’s release model (diffusion regulated) [43]. The kinetics of ME could be explained by the fact that being a lipophilic drug, SLM has a higher partition co-efficient for oil, and its release was hindered at the oil–water interface due to low aqueous solubility. Furthermore, the dialysis bag serves as a barrier to drug release by allowing the free drug to pass through nano-sized perforations [26].

### 3.6. Ex Vivo Diffusion Test

The results of diffusion trials in an ex vivo environment with SLM formulations using sheep nasal mucous membranes are shown in Figure 9. In comparison to MEs, the drug solution (SLMS) demonstrated a considerably decreased flux (*p* < 0.001). Flux for SLMS, SLMME, and SLMMME was found to be 263.18 ± 54.84, 394.56 ± 74.98, and 512.76 ± 96.39 µg/cm^2^ h, respectively. In this study, no statistically considerable difference (*p* > 0.05) was seen among the ME formulations. It was apparent that non-mucoadhesive microemulsions of SLM (SLMME) had a lower flux compared to the mucoadhesive microemulsion (SLMMME), which may be due to infiltration improving the properties of chitosan [44]. In addition to its bio-adhesive and absorption-boosting properties, chitosan has been shown to open tight epithelial junctions in the nasal mucosa, which may account for the greater drug flux [45]. Because the mucus layer in the nasal canal contains a negative charge, it was hypothesized that using chitosan as a mucoadhesive polymer would interact closely with mucus, extending drug residence duration.

### 3.7. In Vivo Pharmacodynamics Studies

#### 3.7.1. Behavioral Factors

##### Locomotor Activity Test

When comparing the disease control group to the control group on the 14th day, it was found that the locomotor activity was considerably (*p* < 0.05) lower in the disease control group. A similar finding was seen on the 21st, 28th, and 35th days (Figure 10). However, on the 35th day, a more significant (*p* < 0.001) decrease in locomotion was observed when compared with the 14th day. No change was observed in the locomotion in the case of the standard drug per se and SLM per se group when compared with the control group on the 0th, 7th, 14th, 21st, 28th, and 35th days. Treatment with standard drugs was not as effective, as a significant (*p* < 0.05) effect was observed only on the 21st day when compared with the rotenone per se group. Treatment with a low dose of SLM microemulsion was found not effective, as no significant effect was observed. However, at a high doses it provided a significant (*p* < 0.05) effect on the 21st, 28th, and 35th day when compared with the rotenone per se group. The effect of a placebo of the mucoadhesive microemulsion was found to be non-significant (*p* > 0.05) when compared with the rotenone per se group on all evaluation days. There was a significant difference (*p* < 0.05) observed when SLM microemulsion was used to treat rotenone poisoning on the 21st, 28th, and 35th days compared to the rotenone alone and placebo groups [26].

##### Muscle Coordination

The muscle coordination significantly (*p* < 0.05) decreased in the rotenone per se group when compared with the control group on the 14th, 21st, 28th, and 35th days (Figure 11). A more significant (*p* < 0.001) decrease in muscle coordination was observed on the 35th day when compared with the 14th day in the same group (rotenone per se group). No significant (*p* > 0.05) change was observed in the standard per se and SLM per se group when compared with the control group. Treatment with standard drugs, naïve SLM, and microemulsion of SLM produced a significant effect only on the 28th and 35th days of evaluation when compared with the control group on the respective day and was more significant (*p* < 0.001) when compared with the same group on the 21st day of evaluation. When compared to the rotenone per se group, the placebo of the mucoadhesive microemulsion was unable to reverse the rotenone effect since no significant (*p* > 0.05) effect was found. The mucoadhesive microemulsion of SLM was found to be effective at both dose levels, as a significant effect was observed on the 21st, 28th, and 35th days of evaluation when compared with the rotenone per se group on the respective day. However, in the case of a high dose of a mucoadhesive microemulsion of SLM, a significant (*p* < 0.05) effect was also observed when compared with the high dose of a microemulsion of SLM on the 28th day, and when compared with standard treatment, naïve SLM and microemulsion of SLM high dose on the 35th day was more significantly (*p* < 0.001) when compared with the 7th day of evaluation [26].

##### Catalepsy

It was observed that no change occurred in the catalepsy in all groups on the 7th day of evaluation. A significant (*p* < 0.05) change was observed in catalepsy in the rotenone per se group on the 14th, 21st, 28th, and 35th day of evaluation when compared with the control group on the respective day and was more significant (*p* < 0.001) on the 35th day when compared with the same group on the 28th day (Figure 12). No change was observed on all evaluation days in the standard per se and SLM per se groups when compared with the control group. A significant (*p* < 0.05) effect was observed when treatment was given with standard drugs, naïve SLM, and microemulsion of SLM on the 28th and 35th day of evaluation when compared with the rotenone per se group on the respective day. The mucoadhesive microemulsion of SLM was found to be more effective as a significant (*p* < 0.05) effect was observed on the 35th day when compared with the rotenone per se, microemulsion of SLM low dose, and placebo group in case of low dose and when compared with rotenone per se, standard drugs, naïve SLM high dose, microemulsion of SLM high dose, and placebo group in case of high dose. There was an even more significant (*p* < 0.001) effect at both dose levels when compared with the same groups on the 28th day [26].

#### 3.7.2. Biochemical Studies

##### Oxidative Parameters

When the estimation of oxidative stress parameters was performed, it revealed that there was no significant (*p* > 0.05) difference in the level of TBARS and nitrite in the case of the standard per se group and SLM per se group when compared with the control group. When matched with the control set, the level of TBARS and nitrites augmented considerably (*p* < 0.05) in the rotenone alone group. When compared to the rotenone alone group, treatment with conventional medicines and naïve SLM resulted in a considerable (*p* < 0.05) decline in the number of nitrites and TBARS. Treatment with microemulsion of SLM also provided beneficial effects, as the level of TBARS and nitrites were significantly (*p* < 0.05) lower when compared with the rotenone per se group as well as when compared with the naive SLM high dose and standard treatment groups. No significant (*p* > 0.05) effect was observed in the group of placebo of mucoadhesive microemulsion when compared with the rotenone per se group [38]. Significant (*p* < 0.05) reduction in the number of nitrites and TBARS was observed when treatment was given with the mucoadhesive microemulsion of SLM at both dose levels when compared with the rotenone alone, standard treatment, naïve SLM, and microemulsion of SLM groups (Table 6).

In addition to the amount of SOD, the CAT, GSH, and antioxidant enzymes were found to be normal in the case of the standard per se and SLM per se groups, as there was no significant (*p* > 0.05) difference observed when compared with the control group. A significant (*p* < 0.05) decrease in the level of antioxidant enzymes was observed in the rotenone per se group when compared with the control group. Treatment with standard drugs and naïve SLM attenuates the effect of rotenone, as there was a significant (*p* < 0.05) difference observed in the level of antioxidant enzymes in these groups when compared with the rotenone per se group. Treatment with microemulsion of SLM also attenuates the effect of rotenone, as a significant (*p* < 0.05) difference was observed when compared with the rotenone per se group and when compared with that of the standard treatment and naïve SLM treatment groups. No significant (*p* > 0.05) difference was observed in the level of antioxidant enzymes in the case of placebo of mucoadhesive formulation group when compared with the rotenone per se group, which represents that the placebo itself has no effect in attenuating the effect of rotenone [38]. It was observed that when treatment was given with a mucoadhesive formulation of SLM, a significant difference (*p* < 0.05) in the level of antioxidant enzymes was observed when compared with the rotenone per se group as well as with standard treatment, naïve SLM high dose, placebo group, and microemulsion of SLM group. The results of oxidative parameters represented that the mucoadhesive microemulsion of SLM attenuates the effect of rotenone in a much better way compared to that of naïve SLM and microemulsion of SLM (Table 6).

##### ELISA Parameters

The results of the ELISA parameters revealed that with the administration of rotenone, the amount of soluble synuclein alpha, BDNF, and dopamine declined, and the level of inflammatory mediators TNF-α and IL-6 increased significantly (*p* < 0.05) in the rotenone per se group when compared with the control group. No significant (*p* > 0.05) change was observed in the level of soluble alpha synuclein, BDNF and dopamine decrease, and level of inflammatory mediators TNF-α and IL-6 in the standard drugs alone groups and naïve SLM high dose group when compared with the control group [38]. Treatment with standard drugs and naïve SLM high dose significantly (*p* < 0.05) reversed the effect of rotenone when compared with the rotenone per se group. It was observed that microemulsion of SLM significantly (*p* < 0.05) reversed the effect of rotenone when compared with the rotenone per se group and standard drugs group. No significant (*p* > 0.05) effect was observed when treatment was given with a placebo of mucoadhesive microemulsion when compared with the rotenone per se group. A significant (*p* < 0.05) effect was observed when treatment was given with a mucoadhesive microemulsion when compared with rotenone per se, standard drugs, SLM high dose, microemulsion of SLM, and placebo of mucoadhesive microemulsion groups (Figure 13, Figure 14, Figure 15, Figure 16 and Figure 17).

## 4. Conclusions

In the present study, mucoadhesive SLMME was developed and optimized using CCD, and further mucoadhesive properties were provided using chitosan. The concentrations of LMCS and T80 were found to be critical in providing the desired droplet size, whereas TP was found effective in providing a desirable zeta potential. With the addition of chitosan, the droplet size and zeta potential of the optimized ME were both increased significantly. In vitro cytotoxicity experiments revealed that the newly developed SLMMME was non-toxic and safe. Mucoadhesive microemulsion demonstrated the greatest flow through sheep nasal mucosa when compared to a microemulsion and drug solution, implying the potential for intranasal delivery of poorly soluble SLM. Further, in vitro release showed significantly higher drug release and diffusion of SLM loaded in MEs than that of SLMS. Further, behavioral and biochemical studies revealed that SLMME and SLMMME delivered considerable enhancement in neuroprotection against PD in a rotenone-induced rat model. Hence, it can be concluded that SLMMME offers a choice for PD treatment as herbal nanomedicine, and proof of concept was provided to initiate clinical studies in the future.

## Figures and Tables

**Figure 1 pharmaceutics-15-00618-f001:**
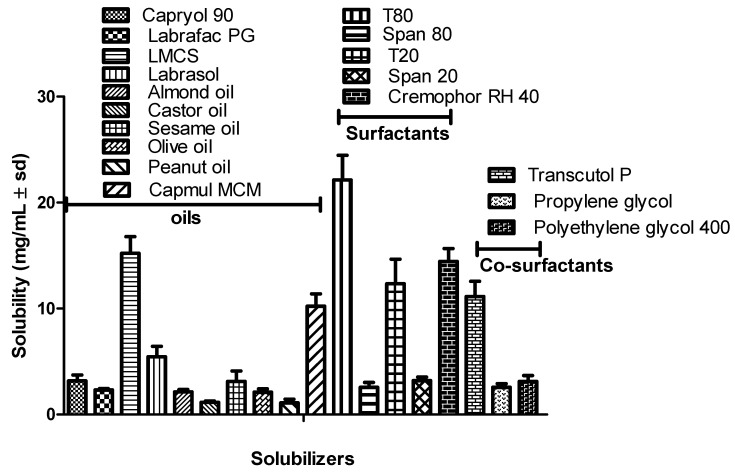
Results of solubility studies of SLM in various solubilizers.

**Figure 2 pharmaceutics-15-00618-f002:**
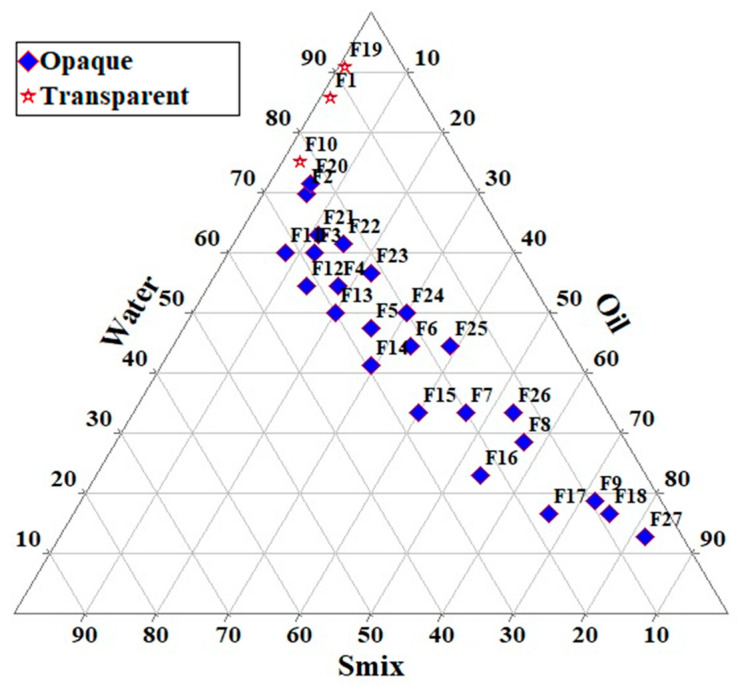
Ternary phase diagram indicating microemulsion region.

**Figure 3 pharmaceutics-15-00618-f003:**
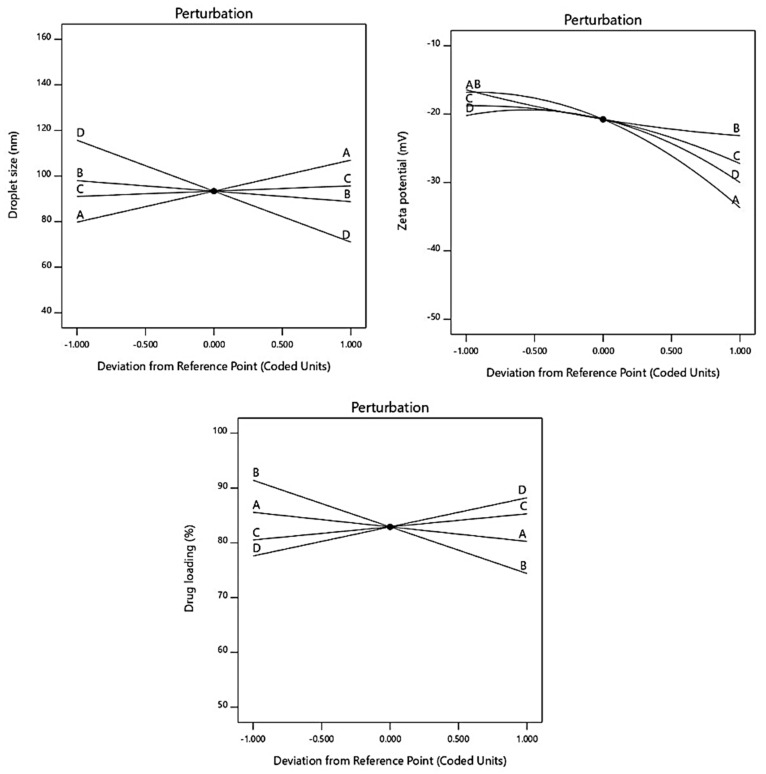
Figure indicating influence of various factors on responses through perturbation graph.

**Figure 4 pharmaceutics-15-00618-f004:**
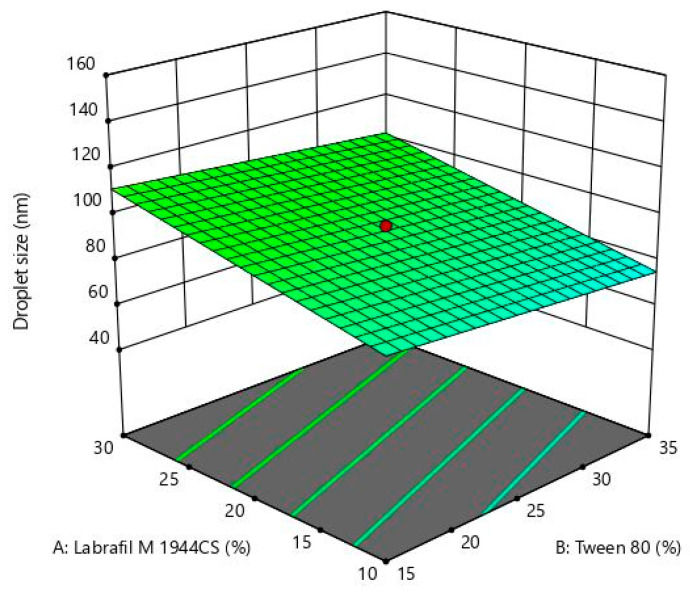
3D response surface plot for Labrafil M 1944CS, Tween 80, Transcutol P, and water on droplet size.

**Figure 5 pharmaceutics-15-00618-f005:**
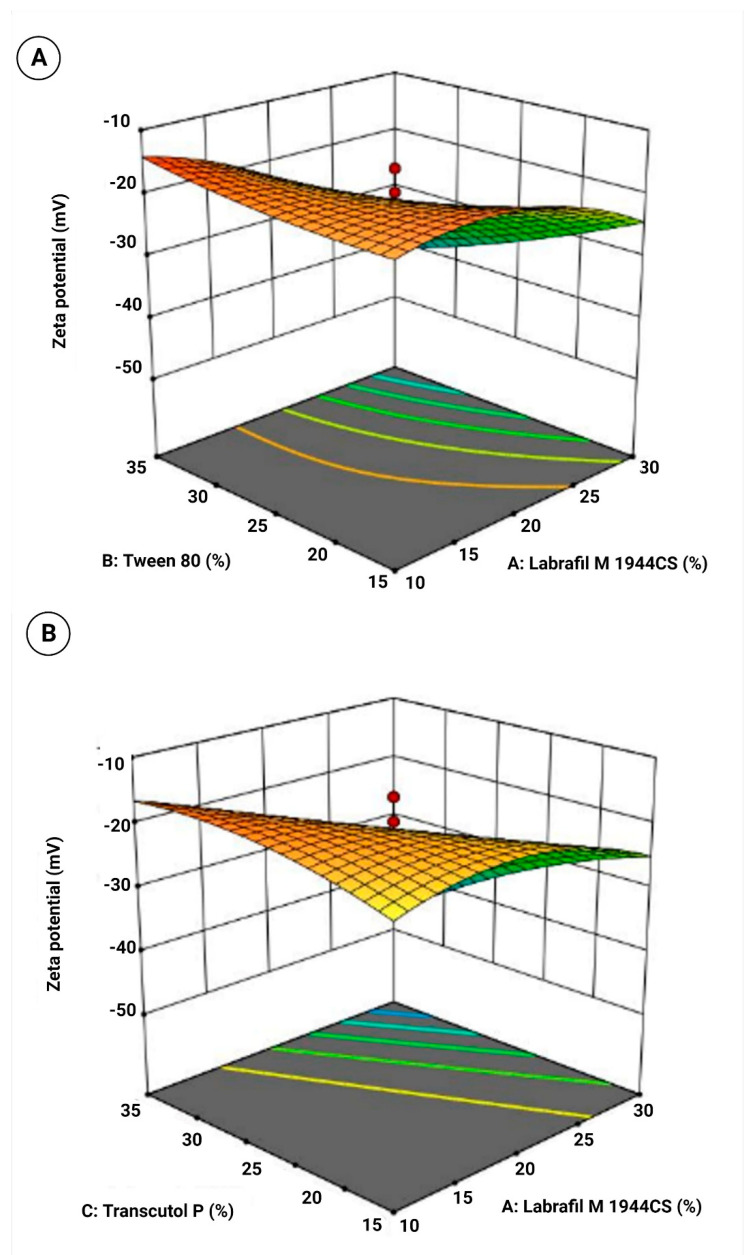
3D response surface plot for (**A**) Labrafil M 1944CS and Tween 80, (**B**) Labrafil M 1944CS and Transcutol P, (**C**) Labrafil M 1944CS and water, (**D**) Tween 80 and Transcutol P, (**E**) Tween 80 and water, and (**F**) Transcutol P and water on zeta potential.

**Figure 6 pharmaceutics-15-00618-f006:**
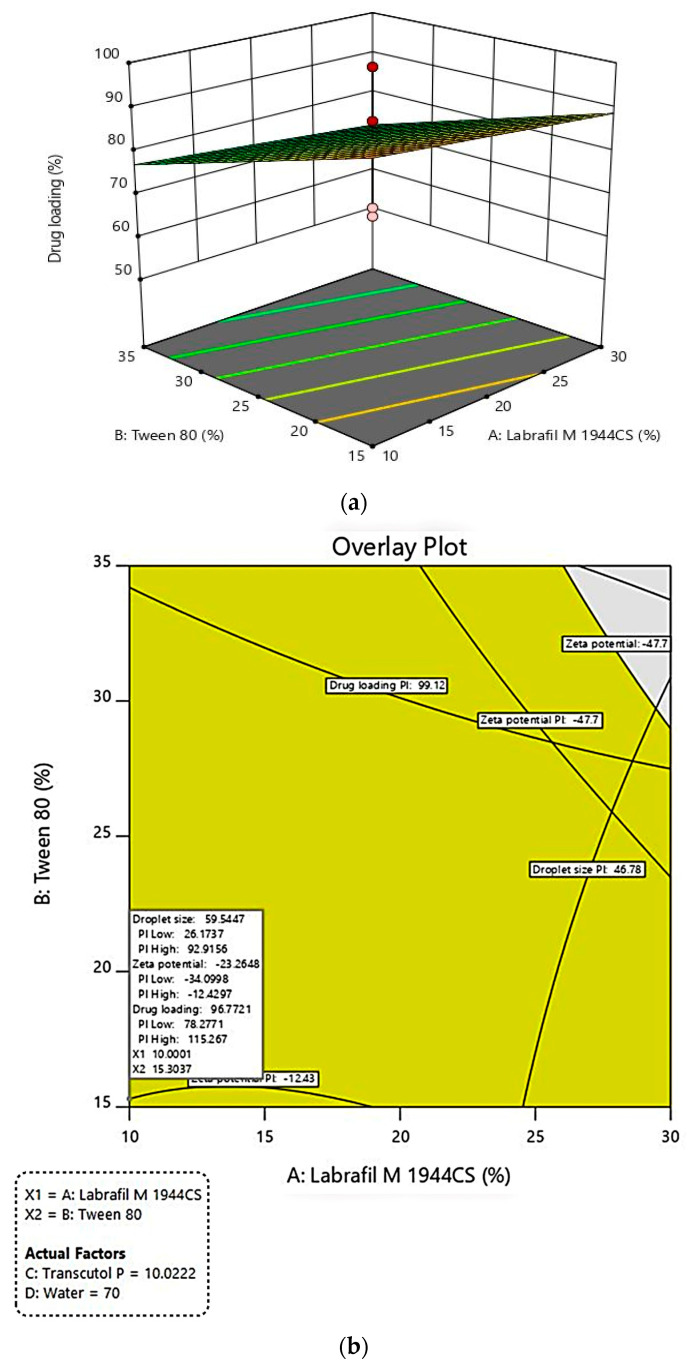
(**a**). 3D response surface plot for effect of formulation variables on drug loading; (**b**). Graphical plot showing predicted values for variables and responses for formulating SLMME.

**Figure 7 pharmaceutics-15-00618-f007:**
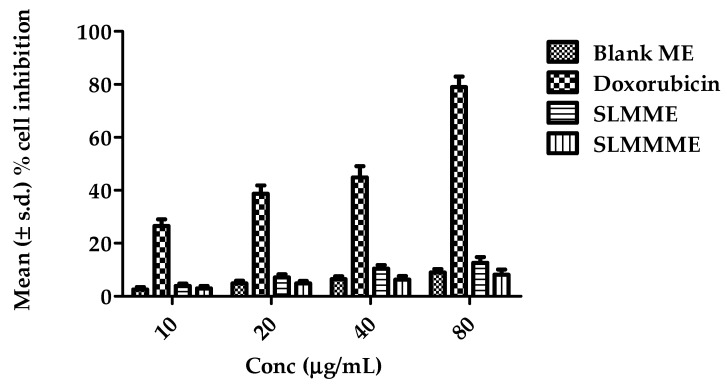
Results of cytotoxicity studies in terms of % cell inhibition of blank ME, Doxorubicin, SLMME, and SLMMME [number of replicates (n) = 3].

**Figure 8 pharmaceutics-15-00618-f008:**
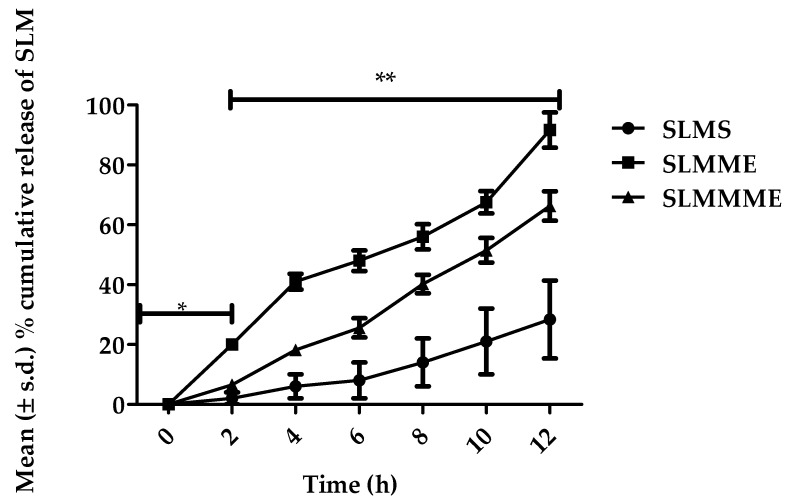
In vitro release profile for SLMME, SLMMME, and SLM drug solutions (SLMCS); * resembles the release profile of SLM in simulated nasal fluid, and ** resembles the release profile of SLM in brain simulated fluid.

**Figure 9 pharmaceutics-15-00618-f009:**
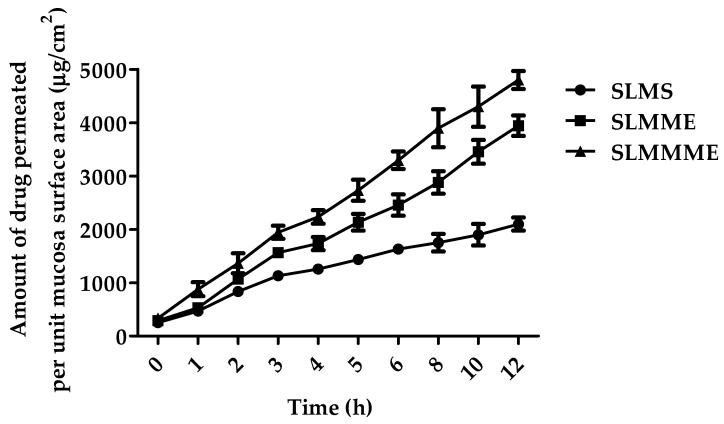
Figure depicting the mean percentage of SLM diffused through sheep nasal mucosa from SLMS, SLMME, and SLMMME.

**Figure 10 pharmaceutics-15-00618-f010:**
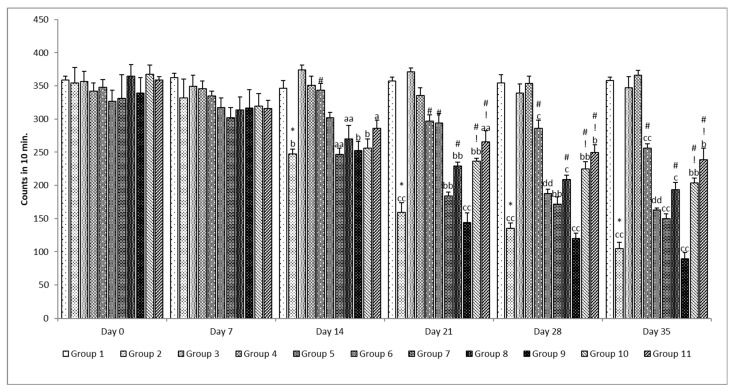
Effect of various treatments on locomotor activity. Note, *, #, and ! represents *p* < 0.05 vs. group 1, 2, 5, 6, 7, 8, 9 and 10. a, b, c, aa, bb, cc and dd represent *p* < 0.05 and *p* < 0.001 vs. day 0, 7, 14, 21 and 28 respectively.

**Figure 11 pharmaceutics-15-00618-f011:**
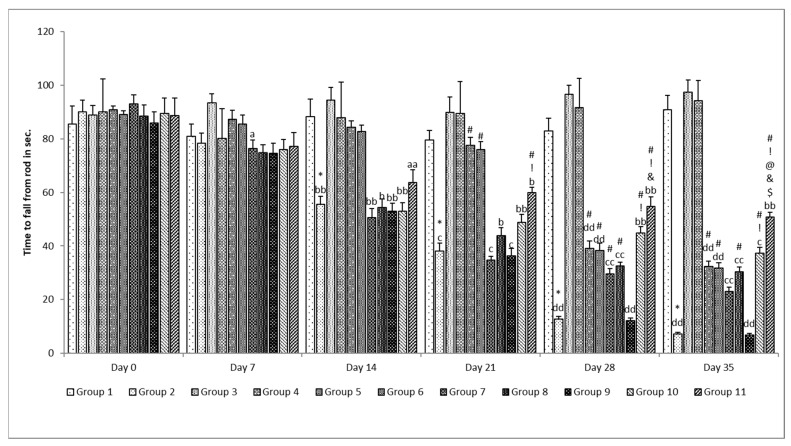
Effect of various treatments on muscle coordination. Note, *, #, @, $, &, and ! represents *p* < 0.05 vs. group 1, 2, 5, 6, 7, 8, 9 and 10. a, b, c, d and aa, bb, cc and dd represent *p* < 0.05 and *p* < 0.001 vs. day 0, 7, 14, 21 and 28 respectively.

**Figure 12 pharmaceutics-15-00618-f012:**
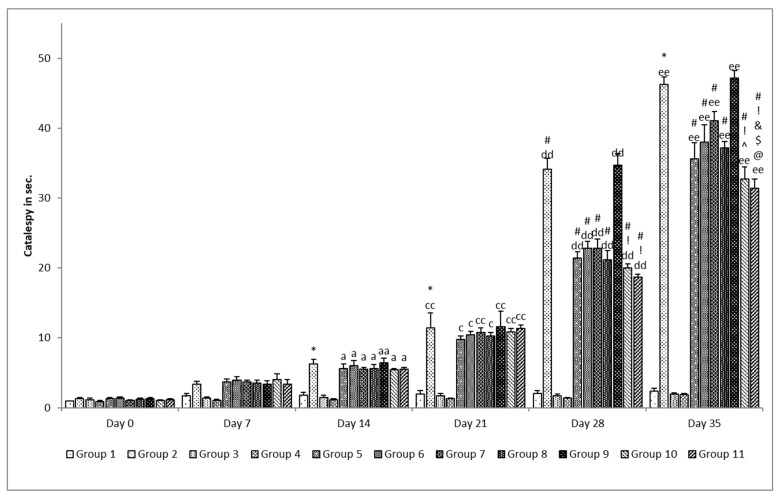
Effect of various treatments on catalepsy in rats. Note, *, #, @, $, ^, &, and ! represents *p* < 0.05 vs. group 1, 2, 5, 6, 7, 8, 9 and 10. a, e and aa, cc, dd and ee represents *p* < 0.05 and *p* < 0.001 vs. day 0, 7, 14, 21 and 28 respectively.

**Figure 13 pharmaceutics-15-00618-f013:**
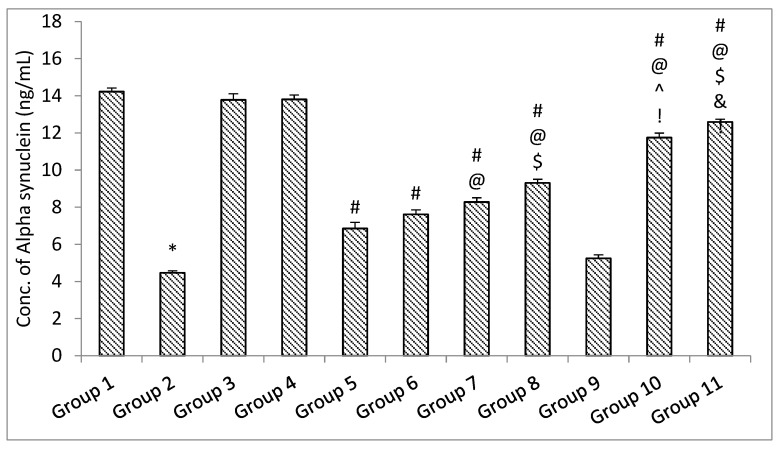
The impact of different treatments on Alfa Synuclein. Data represented as mean ± S.E.M. Note: Data represented as mean ± S.E.M. *, #, @, $, ^, & and ! represent *p* < 0.05 vs. group 1, 2, 5, 6, 7, 8, 9 and 10 respectively.

**Figure 14 pharmaceutics-15-00618-f014:**
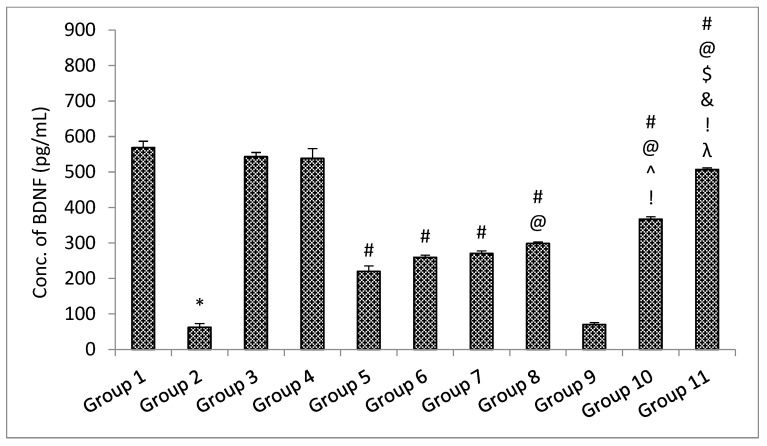
The impact of different treatments on brain-derived neurotropic factor (BDNF). Data represented as mean ± S.E.M. Note: Data represented as mean ± S.E.M. *, #, @, $, ^, &, ! and λ represent *p* < 0.05 vs. group 1, 2, 5, 6, 7, 8, 9 and 10 respectively.

**Figure 15 pharmaceutics-15-00618-f015:**
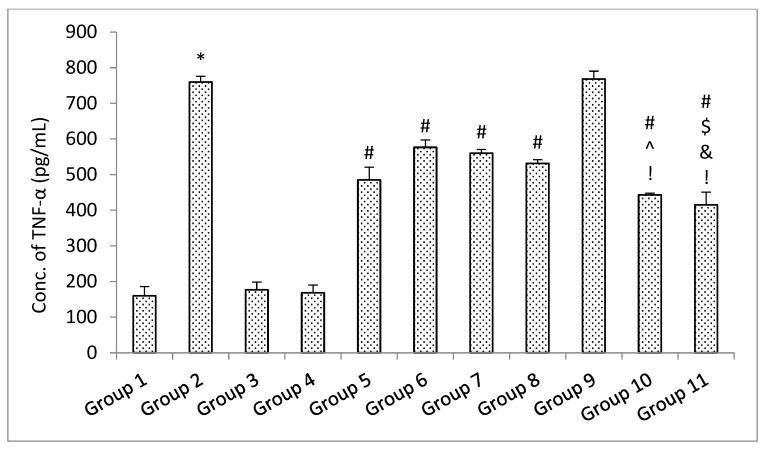
The impact of different treatments on TNF-α. Data represented as mean ± S.E.M. Data represented as mean ± S.E.M. Note: Data represented as mean ± S.E.M. *, #, $, ^, &, and ! represent *p* < 0.05 vs. group 1, 2, 5, 6, 7, 8, 9 and 10 respectively.

**Figure 16 pharmaceutics-15-00618-f016:**
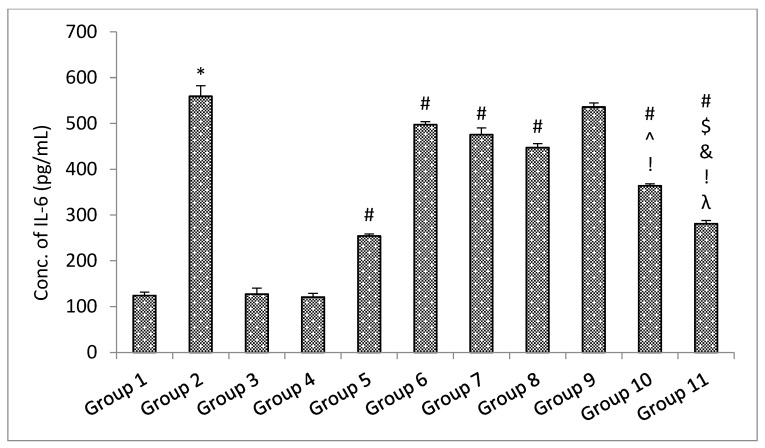
The impact of different treatments on IL-6. Data represented as mean ± S.E.M. Note: Data represented as mean ± S.E.M. *, #, $, ^, &, ! and λ represent *p* < 0.05 vs. group 1, 2, 5, 6, 7, 8, 9 and 10 respectively.

**Figure 17 pharmaceutics-15-00618-f017:**
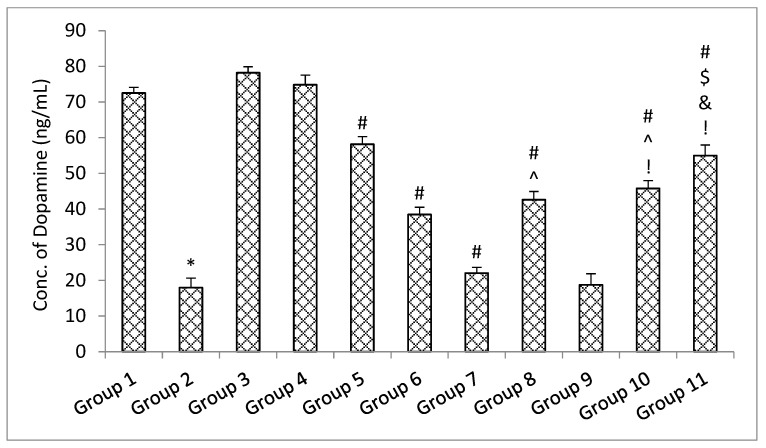
The impact of different treatments on a dopamine. Data represented as mean ± S.E.M. Data represented as mean ± S.E.M. *, #, $, ^, & and ! represent *p* < 0.05 vs. group 1, 2, 5, 6, 7, 8, 9 and 10 respectively.

**Table 1 pharmaceutics-15-00618-t001:** Details of formulation and process variables for optimizing microemulsion.

Independent Factors		
Uncoded (%*v*/*v*,)	Coded	Uncoded
LMCS (Oil phase)	A	10–30
T80 (Surfactant)	B	15–35
TP (Co-surfactant)	C	10–20
Water (Dispersion medium)	D	30–70

**Table 2 pharmaceutics-15-00618-t002:** Protocol of pharmacodynamics’ study.

Groups.	Intervention
Group 1	Control group (mixture of sunflower oil and normal saline)
Group 2	Rotenone alone (disease group)
Group 3	Standard drugs (Levodopa and Carbidopa)
Group 4	Silymarin high dose alone
Group 5	Rotenone + Standard drugs
Group 6	Rotenone + Silymarin HD
Group 7	Rotenone + Silymarin microemulsion (SLMME) LD
Group 8	Rotenone + Silymarin microemulsion (SLMME) HD
Group 9	Rotenone + Placebo mucoadhesive microemulsion
Group 10	Rotenone + Silymarin mucoadhesive microemulsion (SLMMME) LD
Group 11	Rotenone + Silymarin mucoadhesive microemulsion (SLMMME) HD

Note: HD = high dose, LD = low dose, Standard drugs = carbidopa and levodopa.

**Table 3 pharmaceutics-15-00618-t003:** CCD-guided factors, and their levels, affecting responses for optimization of microemulsion.

Run	Factor 1: Labrafil M 1944CS (%*v*/*v*,)	Factor 2: Tween 80 (%*v*/*v*,)	Factor 3: Transcutol P (%*v*/*v*,)	Factor 4: Water (%*v*/*v*,)	Y_1_: Droplet Size (nm)	Y_2_: Zeta Potential (mV)	Y_3_: Drug Loading (%)
1	10.00	15	20.00	30.00	99.24	−29.7	97.99
2	30.00	15	20.00	70.00	74.53	−41.8	99.01
3	20.00	25	6.59	50.00	69.09	−18.4	84.01
4	20.00	25	15.00	50.00	73.70	−15.91	65.02
5	30.00	15	10.00	70.00	92.98	−27.9	90.18
6	20.00	25	15.00	50.00	95.51	−22.8	87.02
7	20.00	41.81	15.00	50.00	96.77	−23.7	57.35
8	30.00	35	20.00	30.00	126.57	−44.67	73.36
9	20.00	8.18	15.00	50.00	110.6	−12.43	87.1
10	20.00	25	15.00	16.36	152.71	−24.87	71.31
11	20.00	25	15.00	50.00	87.93	−23.67	81.58
12	20.00	25	15.00	83.63	88.10	−41.34	94.65
13	36.81	25	15.00	50.00	116.73	−47.7	86.34
14	20.00	25	15.00	50.00	71.81	−21.67	99.12
15	10.00	35	20.00	70.00	46.78	−21.43	82.9
16	20.00	25	23.40	50.00	114.33	−35.9	86.9
17	10.00	35	10.00	70.00	56.68	−31.87	83.99
18	20.00	25	15.00	50.00	85.85	−19.65	67.03
19	3.18	25	15.00	50.00	58.86	−19.3	94.99
20	10.00	15	10.00	30.00	122.34	−32.1	85.36
21	30.00	35	10.00	30.00	119.49	−17.4	65.97

**Table 4 pharmaceutics-15-00618-t004:** ANOVA summary for responses pertaining to the variables of CCD.

Response Variables	Parameters of Regression		*p* Value
	R^2^	Fcal.	
Droplet size (Y1)	0.6895	8.88	0.0006
Zeta potential (Y2)	0.9765	17.83	0.0010
Drug loading	0.5358	4.62	0.0114

**Table 5 pharmaceutics-15-00618-t005:** Results of release kinetics models applied for ME.

Model	SLMME	SLMMME
	R^2^ value	R^2^ value
Zero order	0.9182	0.8256
First order	0.4183	0.3460
Korsmeyer Peppas	0.9502	0.9188
Weibull	0.9195	0.9089
Hixon Crowell	0.7262	0.6637
Higuchi	0.9996	0.9920

**Table 6 pharmaceutics-15-00618-t006:** Data representing effect of varying treatments on oxidative stress parameters.

Treatment	Oxidative Parameter				
	TBARS (nmol/mg of Protein)	Nitrite (nmol/mg of Protein)	GSH (nmol/mg of Protein)	CAT (Units/mg of Protein)	SOD (Units/mg of Protein)
Group 1	1.117 ± 0.028	0.807 ± 0.012	4.452 ± 0.122	0.975 ± 0.016	11.444 ± 0.311
Group 2	3.530 ± 0.313 *	1.777 ± 0.029 *	0.955 ± 0.050 *	0.057 ± 0.023 *	3.210 ± 0.781 *
Group 3	1.180± 0.020	0.789 ± 0.006	4.465 ± 0.164	0.953 ± 0.034	11.649 ± 0.334
Group 4	1.178 ± 0.042	0.806 ± 0.012	4.283 ± 0.119	0.926 ± 0.022	11.611 ± 0.405
Group 5	2.901 ± 0.030 ^#^	1.498 ± 0.018 ^#^	2.145 ± 0.074 ^#^	0.420 ± 0.043 ^#^	5.292 ± 0.537 ^#^
Group 6	2.708 ± 0.030 ^#^	1.410 ± 0.018 ^#^	2.445 ± 0.074 ^#^	0.461 ± 0.040 ^#^	5.872 ± 0.496 ^#^
Group 7	2.592 ± 0.038 ^#^	1.472 ± 0.024 ^#^	2.985 ± 0.033 ^#,@^	0.554 ± 0.045 ^#^	6.411 ± 0.449 ^#^
Group 8	1.910 ± 0.032 ^#,@,$,^^	1.333 ± 0.010 ^#,@,^^	3.461 ± 0.030 ^#,@,$^	0.629 ± 0.031 ^#,$^	6.996 ± 0.749 ^#^
Group 9	3.525 ± 0.313	1.677 ± 0.029	0.915 ± 0.050	0.052 ± 0.026	3.310 ± 0.740
Group 10	1.761 ± 0.073 ^#,@,^,!^	1.339 ± 0.020 ^#,@,^,!^	2.740 ± 0.005 ^#,@,^,!^	0.889 ± 0.014 ^#,@,^,!^	9.508 ± 0.380 ^#,@,^,!^
Group 11	1.228 ± 0.108 ^#,@,$,&,!^	1.149 ± 0.025 ^#,@,$,&,!,λ^	3.921 ± 0.231 ^#,@,$,&,!,λ^	0.910 ± 0.023 ^#,@,$,&,!^	10.416 ± 0.634 ^#,@,$,&,!^

Note: Data represented as mean ± S.E.M. *, ^#^, ^@^, ^$^, ^^^, ^&^, ^!^, and ^λ^ represent *p* < 0.05 vs. group 1, 2, 5, 6, 7, 8, 9, and 10, respectively.

## Data Availability

Not applicable.

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
