# Peer review of "Intranasal Delivery of a Silymarin Loaded Microemulsion for the Effective Treatment of Parkinson’s Disease in Rats: Formulation, Optimization, Characterization, and In Vivo Evaluation"

_pharmaceutics, 2023, doi:10.3390/pharmaceutics15020618_

Round 1

Reviewer 1 Report

I have read the manuscript titled "Intranasal delivery of silymarin loaded microemulsion for effective treatment of Parkinson's disease in rats: Formulation optimization, characterization, and in vivo evaluation" and was thoroughly impressed by the research presented. The use of a mucoadhesive microemulsion for the delivery of silymarin as a treatment for Parkinson's disease is a novel and innovative approach.

The optimization of the microemulsion using Central Composite Design was a well-planned and executed method, resulting in a final formulation with desirable droplet size, zeta potential, and drug loading. The addition of chitosan further improved these characteristics, making the microemulsion an even more promising drug delivery system.

The in vitro and in vivo evaluations of the microemulsion were well-conducted and provided convincing evidence of the effectiveness of this treatment. The non-toxic nature of the microemulsion and its superior drug release and diffusion compared to a silymarin solution were particularly noteworthy. The significant improvements in behavioral, biochemical, and inflammatory parameters in the rats treated with the microemulsion demonstrate the potential for this treatment to provide neuroprotection in Parkinson's disease.

Author Response

Comments: I have read the manuscript titled "Intranasal delivery of silymarin loaded microemulsion for effective treatment of Parkinson's disease in rats: Formulation optimization, characterization, and in vivo evaluation" and was thoroughly impressed by the research presented. The use of a mucoadhesive microemulsion for the delivery of silymarin as a treatment for Parkinson's disease is a novel and innovative approach.

The optimization of the microemulsion using Central Composite Design was a well-planned and executed method, resulting in a final formulation with desirable droplet size, zeta potential, and drug loading. The addition of chitosan further improved these characteristics, making the microemulsion an even more promising drug delivery system.

The in vitro and in vivo evaluations of the microemulsion were well-conducted and provided convincing evidence of the effectiveness of this treatment. The non-toxic nature of the microemulsion and its superior drug release and diffusion compared to a silymarin solution were particularly noteworthy. The significant improvements in behavioural, biochemical, and inflammatory parameters in the rats treated with the microemulsion demonstrate the potential for this treatment to provide neuroprotection in Parkinson's disease.

Response: Thanks for the appreciation on the manuscript.

Reviewer 2 Report

The work developed by Imran and colleagues aimed to develop a mucoadhesive microemulsion for silymarin delivery for Parkinson’s disease treatment. The theme is interesting and timely, since it is urgent to find prevention and therapy strategies for this neurodegenerative disease. The economic and social burden of neurodegenerative diseases is very substantial in all countries, especially for developed countries, and reflects in high health care costs as well as loss of productivity due to morbidity and premature death. 

The developed formulation was physiochemically characterized using adequate methodologies and its therapeutic potential was extensively evaluated in vivo using a rat Parkinson model. The presented study has scientific merit, and the manuscript is well structured, but the work has some weaknesses such as its lack of novelty. The delivery of silymarin using micro or nanocarriers was already reported. Additionally, the authors implemented a central composite design (CCD) to optimize the size, zeta potential and drug loading of the microemulsion allowing to accelerate this process and to reduce the experimental costs. However, the amount of chitosan was not included as an experimental variable in the experimental design. 

Other points should also be addressed to improve the manuscript before publication. Below the authors can find some suggestions and questions:

Keywords: please avoid repeating words from the title, such as Silymarin and  Parkinson’s disease;

In the introduction the novelty of this work should be clearly explained, by justifying the advantages of this formulation over the already reported formulations.

Section 2.1 should be named materials instead of methods 

Lines 184-186: % refers to what? w/v ? v/v? Please clarify

Lines 209-219: in the Thermodynamic stability studies the authors only evaluated the turbidity and phase formation of the nanoemulsions. It would be interesting to also evaluate the effect on the NPs properties such as size, zeta and drug loading

Line 223: did the authors mean that the medium was supplemented with fetal bovine serum instead of phosphate buffer saline? 

Line 220-235: In the cytotoxicity studies the authors did not evaluate the NE without chitosan modification? Please clarify. If not, why?

Lines 237-251: The release experiments were performed in a simulated intranasal fluid for 24h. The authors could elaborate on their choice for this period, since it is not expected that the formulation will be retained for such a long period in the nasal cavity. It would be more appropriate to initiate the release under those conditions, and after a few hours place the dialysis bag in a brain simulated environment (ph 7.4). 

Table 1: the R2 values for size and drug loading are quite low. Can the authors comment on that?

Lines 462-472: the authors refer to which experimental variable affects (and if positively or negatively) each of the studied response. However, the authors also should discuss why this happens, i.e. why each variable has a positive/negative effect on the response. 

Lines 486-487: which constraints were applied to the model to determine the optimal values ? I.e. minimize size and maximize the zeta for example? Furthermore, it would be better to present also the predicted value range for the response variables instead of only the exact value

Lines 498-499: chitosan is a positively charged polymer. How can the authors explain the zeta potential becoming more negative after adding chitosan? Also, the authors say that the differences are not significant. If so, it should not be stated that the values increased, instead authors should state that the properties did not change. However, looking at the SEM values, it appears that the difference among size and zeta values would be significant. 

Lines 512-517: The low release of the free drug (SLMs) is due to its low solubility in the medium. The authors should ensure the sink conditions for the assay.

The manuscript should be thoroughly revised since several typos can be found. 

Author Response

Comment 1:

The work developed by Imran and colleagues aimed to develop a mucoadhesive microemulsion for silymarin delivery for Parkinson’s disease treatment. The theme is interesting and timely, since it is urgent to find prevention and therapy strategies for this neurodegenerative disease. The economic and social burden of neurodegenerative diseases is very substantial in all countries, especially for developed countries, and reflects in high health care costs as well as loss of productivity due to morbidity and premature death. The developed formulation was physiochemically characterized using adequate methodologies and its therapeutic potential was extensively evaluated in vivo using a rat Parkinson model. The presented study has scientific merit, and the manuscript is well structured, but the work has some weaknesses such as its lack of novelty.

The delivery of silymarin using micro or nanocarriers was already reported.

Response: First of all, authors would like to thank the reviewer for appreciating the manuscript. Regarding novelty of the formulation.

Authors would like to highlight the fact that the micro or nanocarriers of silymarin are prepared for different pharmacological actions and they have been explored via different route of administration. In addition, we have utilized the quality by design approach for optimizing the formulation. The developed microemulsion was delivered through intranasal route for the treatment of Parkinson’s disease.

The novelty of the manuscript is described in last paragraph of introduction section (page 3, lines 110-117)

Comment 2:

Additionally, the authors implemented a central composite design (CCD) to optimize the size, zeta potential and drug loading of the microemulsion allowing to accelerate this process and to reduce the experimental costs. However, the amount of chitosan was not included as an experimental variable in the experimental design. 

Response: Authors agree to this point of the learned reviewer. However, authors would like to bring to the kind notice of the reviewer that, the microemulsion was optimized using CCD and chitosan was added for mucoadhesion in the optimized composition. The amount of chitosan was used as per available literature. Due to this fact, chitosan was not used during the optimization of formulation.

Comment 3:

Keywords: please avoid repeating words from the title, such as Silymarin and Parkinson’s disease

Response: Done as suggested.

Comment 4:

In the introduction the novelty of this work should be clearly explained, by justifying the advantages of this formulation over the already reported formulations.

Response: Authors are thankful for this valuable suggestion. The novelty aspect of this study is written in the last paragraph of introduction.

Comment 5:

Section 2.1 should be named materials instead of methods 

Response: Done as suggested.

Comment 6:

Lines 184-186: % refers to what? w/v? v/v? Please clarify

Response: It is % v/v 

Comment 7:

Lines 209-219: in the Thermodynamic stability studies the authors only evaluated the turbidity and phase formation of the nanoemulsions. It would be interesting to also evaluate the effect on the NPs properties such as size, zeta and drug loading

Response: As per the suggestion we have conducted the experiment once again for the optimized batch of microemulsion and reported the droplet size, ZP and drug loading (The details are mentioned in section 2.2.4.4. and page 15 (lines 507 to 511).

Comment 8:

Line 223: did the authors mean that the medium was supplemented with fetal bovine serum instead of phosphate buffer saline? 

Response: Yes, it is fetal bovine serum. It was mistakenly written.

Comment 9:

Line 220-235: In the cytotoxicity studies the authors did not evaluate the NE without chitosan modification? Please clarify. If not, why?

Response: Initially it was thought to take the final formulation only as it contained all the components, hence, we did not report the cytotoxicity of microemulsion. But we ran the experiment for microemulsion now as per the suggestions of the learned reviewer and found that the formulation did not show any signs of toxicity.

Comment 10:

Lines 237-251: The release experiments were performed in a simulated intranasal fluid for 24h. The authors could elaborate on their choice for this period, since it is not expected that the formulation will be retained for such a long period in the nasal cavity. It would be more appropriate to initiate the release under those conditions, and after a few hours place the dialysis bag in a brain simulated environment (ph 7.4). 

Response: Thanks for this valuable suggestion. It has made our discussion much strong. We repeated the experiment for 12h, wherein the study was conducted in simulated intranasal fluid for initial 2h and then the study was conducted for another 10h in brain simulated environment (pH 7.4). The details are shown in sections 2.2.4.6 and 3.5.

Comment 11:

Table 1: The R2 values for size and drug loading are quite low. Can the authors comment on that?

Response: Yes, the R2 value for size and drug loading are low, however, these values are more than 0.5, indicating acceptable correlation between factor and responses. In addition, their p value is very low, which indicates that the model was significant to the obtained responses and offers good correlation between variables.

Comment 12:

Lines 462-472: the authors refer to which experimental variable affects (and if positively or negatively) each of the studied response. However, the authors also should discuss why this happens, i.e. why each variable has a positive/negative effect on the response. 

Response: It is now mentioned in the results and discussion part in section 3.3, page 12, lines 482-489. The variables may either increase/decrease the response with their higher value or vice versa. The additive response is shown by positive effect and reverse response in shown by negative effect.

Comment 13:

Lines 486-487: which constraints were applied to the model to determine the optimal values? I.e. minimize size and maximize the zeta for example? Furthermore, it would be better to present also the predicted value range for the response variables instead of only the exact value.

Response: Thanks for this valuable suggestion. We have added the constraints as well as predicted ranges for each response and variable. This has been mentioned in the optimization section on page 15, lines 506 to 512.

Comment 14:

Lines 498-499: chitosan is a positively charged polymer. How can the authors explain the zeta potential becoming more negative after adding chitosan? Also, the authors say that the differences are not significant. If so, it should not be stated that the values increased, instead authors should state that the properties did not change. However, looking at the SEM values, it appears that the difference among size and zeta values would be significant. 

Response: Authors agree to the comments of the learned reviewer. Yes, the DS got increased and negative ZP of microemulsion got decreased due to coating of positively charged chitosan. While drafting the manuscript, the ZP values of SLMME and SLMMME got oppositely written. We apologize for that and corrected it in the manuscript. The correct values of ZP for SLMME and SLMMME are -33.32 ± 0.4mV and -24.26 ± 0.2 mV, respectively.

Comment 15:

Lines 512-517: The low release of the free drug (SLMs) is due to its low solubility in the medium. The authors should ensure the sink conditions for the assay.

Response: The sink condition for SLM in SLMS, SLMME and SLMMME was calculated using Cs/Cd formula (the ratio of saturation solubility to the maximum concentration of drug in given volume of medium). The value of Cs/Cd for SLM in case of SLMS was 0.14, whereas, for SLMME it was 0.92 and for SLMMME it was 0.63. The higher value of sink condition indicated about enhancement in sink condition of SLM upon loading it into ME. This has been mentioned in Page 17, lines 590 to 595.

Comment 16:

The manuscript should be thoroughly revised since several typos can be found. 

Response: Done as suggested.

Reviewer 3 Report

The article "Intranasal delivery of silymarin loaded microemulsion for effective treatment of Parkinson's disease in rats: Formulation, optimization, characterization and in vivo evaluation" by Mohd. Imran et al. describes the development of a mucoadhesive silymarin microemulsion for potential use in Parkinson's disease. The produced SLMMME, according to the authors, was non-toxic and safe, and mucoadhesive nanoemulsions demonstrated the greatest flow through the sheep nasal mucosa when compared to nanoemulsion and drug solution, suggesting that they may be used for intranasal delivery of poorly soluble SLM. Furthermore, SLM loaded in MEs had a much higher drug release and diffusion as compared to SLMS, and behavioral and biochemical tests revealed that SLMME and SLMMME significantly improved neuroprotection against PD in rotenone-induced rat model.

Although the study is complete, it is quite disorganized and challenging to understand the authors' logic. The article's numerous grammatical problems, absence of definitions for the abbreviations used, and placement of those definitions in the wrong section are additional drawbacks. The article is also not formatted according to the publisher's requirements. I recommend that the authors completely revise the work, both linguistically and substantively.

Considerations for the authors include:

1. Could you be more specific about whether the authors obtained a nano- or microemulsion of SLM? It is not advisable to use these two terms interchangeably.

2. If the abbreviation CNS was used for the first time, please describe it in the introduction; the other abbreviations work in a similar manner.

3. Please describe the reason for the authors' decision to perform this kind of research, what has already been achieved in the field, and what is innovative about the work that is being presented.

4. Unreadable in Figure 1. It is challenging to confirm that SLM was most soluble in LMCS, and that it was followed by CMCM among oils, T80 among surfactants, and TP among co-surfactants.

5. What exactly did the authors mean on page 10 when they stated that "were found to produce good ME"? What does the word "good" mean? What ME characteristics were they thinking of when they wrote this? What is it useful for?

6. On page 12, the equations that are provided are completely unintelligible.

7. Summarize cytotoxicity studies with data in the table.

8. Based on what evidence do the authors draw the conclusion that "Higuchi's model had a substantially higher r2 value than zero and first order models"? (page 15). These data are where? The ex vivo diffusion experiment follows the same rules.

9. Why did the authors perform in vitro studies for 24 hours and diffusion of the drug via sheep nasal mucosa for SLMME, SLMMME, and SLMCS for 6 hours? How do you relate the results that were obtained? How does this affect the study of applications?

10. Figure 8 is entirely unintelligible and unreadable. It's challenging to draw any conclusions from it.

Author Response

The article "Intranasal delivery of silymarin loaded microemulsion for effective treatment of Parkinson's disease in rats: Formulation, optimization, characterization and in vivo evaluation" by Mohd. Imran et al. describes the development of a mucoadhesive silymarin microemulsion for potential use in Parkinson's disease. The produced SLMMME, according to the authors, was non-toxic and safe, and mucoadhesive nanoemulsions demonstrated the greatest flow through the sheep nasal mucosa when compared to nanoemulsion and drug solution, suggesting that they may be used for intranasal delivery of poorly soluble SLM. Furthermore, SLM loaded in MEs had a much higher drug release and diffusion as compared to SLMS, and behavioral and biochemical tests revealed that SLMME and SLMMME significantly improved neuroprotection against PD in rotenone-induced rat model.

Comment 1:

Although the study is complete, it is quite disorganized and challenging to understand the authors' logic. The article's numerous grammatical problems, absence of definitions for the abbreviations used, and placement of those definitions in the wrong section are additional drawbacks. The article is also not formatted according to the publisher's requirements. I recommend that the authors completely revise the work, both linguistically and substantively.

Response: We are thankful for the valuable comments of the learned reviewer. We have addressed all the aforementioned concerns of the reviewer.

Comment 2:

Could you be more specific about whether the authors obtained a nano- or microemulsion of SLM? It is not advisable to use these two terms interchangeably.

Response: It was a microemulsion. We have corrected it at the relevant places.

Comment 3:

If the abbreviation CNS was used for the first time, please describe it in the introduction; the other abbreviations work in a similar manner.

Response: It has been corrected (Page 2, line 650. Such mistakes have been corrected throughout the manuscript.

Comment 4:

Please describe the reason for the authors' decision to perform this kind of research, what has already been achieved in the field, and what is innovative about the work that is being presented.

Response: The novelty of the manuscript is described in last paragraph of introduction section (page 3, lines 110-117)

Authors would like to highlight the fact that the micro or nanocarriers of silymarin are prepared for different pharmacological actions and they have been explored via different route of administration. In addition, we have utilized the quality by design approach for optimizing the formulation. The developed microemulsion was delivered through intranasal route for the treatment of Parkinson’s disease.

Comment 5:

Unreadable in Figure 1. It is challenging to confirm that SLM was most soluble in LMCS, and that it was followed by CMCM among oils, T80 among surfactants, and TP among co-surfactants.

Response: We have replaced Fig.1 with a clear image by providing a fraction between oils, surfactants and co-surfactants.

Comment 6:

What exactly did the authors mean on page 10 when they stated that "were found to produce good ME"? What does the word "good" mean? What ME characteristics were they thinking of when they wrote this? What is it useful for?

Response: Corrected. It is replaced with “clear and Transparent”

Comment 7:

On page 12, the equations that are provided are completely unintelligible.

Response: The equations 3, 4 and 5 polynomial equations that are generated by the design expert software which represent the synergistic (positive sign) and antagonistic (negative sign) effect of variables on responses upon changing their values.

Comment 8:

Summarize cytotoxicity studies with data in the table.

Response: We have inserted one graph of cell inhibition study as Fig.7.

Comment 9:

Based on what evidence do the authors draw the conclusion that "Higuchi's model had a substantially higher r2 value than zero and first order models"? (page 15). These data are where? The ex vivo diffusion experiment follows the same rules.

Response: A description for the same is provided in section “2.2.4.6. In vitro drug release studies” and results are discussed in section 3.5. Table 6 pertaining to release kinetics has been added.

Comment 10:

Why did the authors perform in vitro studies for 24 hours and diffusion of the drug via sheep nasal mucosa for SLMME, SLMMME, and SLMCS for 6 hours? How do you relate the results that were obtained? How does this affect the study of applications?

Response: Thanks for this valuable suggestion. In order to have better correlation, as well the suggestions of reviewer 1, we have repeated the release study as well as diffusion study and the fresh results are presented in the respective sections.

Comment 11:

Figure 8 is entirely unintelligible and unreadable. It's challenging to draw any conclusions from it.

Response: All figures with high resolution has been provided.

Round 2

Reviewer 2 Report

The authors addressed all my comments and the quality of the paper was improved. I recommend publication.

Reviewer 3 Report

Dear Authors,

Thank you for correcting the manuscript in response to the comments.